# COVI-Prim survey: Challenges for Austrian and German general practitioners during initial phase of COVID-19

Andrea Siebenhofer[1,2]☯, Sebastian Huter[3]☯, Alexander Avian [4]*, Karola Mergenthal[2], Dagmar Schaffler-Schaden[3], Ulrike Spary-Kainz[1], Herbert Bachler[5], Maria Flamm[3]

1 Institute of General Practice and Evidence based Health Services Research, Medical University Graz, Graz, Austria, 2 Institute of General Practice, Johann Wolfgang Goethe University Frankfurt, Frankfurt, Germany, 3 Institute for General Practice, Family Medicine and Preventive Medicine, Paracelsus Medical University Salzburg, Salzburg, Austria, 4 Institute for Medical Informatics, and Statistics and Documentation, Medical University Graz, Graz, Austria, 5 Institute of General Practice, Medical University Innsbruck, Innsbruck, Austria

☯ These authors contributed equally to this work.
* alexander.avian@medunigraz.at

## Abstract

### Background

Coronavirus disease 2019 (COVID-19) represents a significant challenge to health care systems around the world. A well-functioning primary care system is crucial in epidemic situations as it plays an important role in the development of a system-wide response.

### Methods

2,187 Austrian and German GPs answered an internet survey on preparedness, testing, staff protection, perception of risk, self-confidence, a decrease in the number of patient contacts, and efforts to control the spread of the virus in the practice during the early phase of the COVID-pandemic (3rd to 30th April).

### Results

The completion rate of the questionnaire was high (90.9%). GPs gave low ratings to their preparedness for a pandemic, testing of suspected cases and efforts to protect staff. The provision of information to GPs and the perception of risk were rated as moderate. On the other hand, the participants rated their self-confidence, a decrease in patient contacts and their efforts to control the spread of the disease highly.

### Conclusion

Primary care is an important resource for dealing with a pandemic like COVID-19. The workforce is confident and willing to take an active role, but needs to be provided with the appropriate surrounding conditions. This will require that certain conditions are met.

**Data Availability Statement:** The authors have uploaded an additional minimal data set to Data

Archiving and Networked Services (DANS) which is available at https://doi.org/10.17026/dans-z8u-unbv.

**Funding:** The authors received no specific funding for this work.

**Competing interests:** The authors have declared that no competing interests exist.

## Registration

Trial registration at the German Clinical Trials Register: DRKS00021231.

## Introduction

Coronavirus disease 2019 (COVID-19) represents a significant challenge to health care systems around the world. Although implications for the hospital and intensive care sector are generally focused on, a comprehensive approach to managing the COVID-19 pandemic should also involve primary care, as it is usually the point-of-first-contact, regardless of patients'health concerns [1, 2]. In a pandemic, it is therefore particularly important that primary care is in a position to provide the continuous care that is needed, especially when other parts of the system are overwhelmed [3].

Primary care professionals represent the first point of contact in health care systems and are therefore in a vulnerable position. With sometimes insufficient information, they must deal with a dilemma between caring for potentially infectious patients [4], while protecting themselves and those around them from contracting the disease [5, 6]. Previous studies have emphasized the need to include general practitioners in preparedness planning and in supplying them with the personal protective equipment (PPE) they require to quickly adapt to highly dynamic epidemiological developments [7, 8]. While scenarios comparable to the COVID-19 pandemic have been simulated [9], national response plans in many countries still tend to neglect the primary care sector [10]. Furthermore, primary care in Austria and Germany is mostly delivered in small, decentralized units run by self-employed general practitioners (GPs), which may hinder a rapid and coordinated pandemic response [11].

Neither Germany nor Austria have yet exhausted their intensive care capacities and have managed to keep infection numbers under control [12, 13]. Nevertheless, it remains unclear how long the COVID-19 pandemic will last. Primary care will likely have to deal with recurring waves of infections, at least in certain regions, especially since dealing with viral infections is part of the daily business of general practice [14].

The aim of this study is to investigate the role played by GPs in the early phase of the COVID-19 pandemic, the specific challenges faced by them, their concerns and the strategies they have developed to cope with the pandemic. Potential deficiencies as well as regional differences (country-specific, setting, urbanity) are analyzed.

## Methods

This manuscript was prepared in accordance with the CHERRIES criteria [15] (Supporting Information A-13). COVI-Prim-*Start* is part of the international COVI-Prim project [16]. Since this is the first publication to emerge from the project, the methods and design of the study are described in detail in the Supplement.

### Questionnaire development

To create a basic pool of items for the COVI-Prim questionnaire, we searched the literature for studies investigating the role of general practice during pandemics. Various topics, which had been partially grouped in topic areas in the literature, were identified. New topic areas were created for topics that did not belong in those found in the literature. Based on the literature review, semi-structured telephone interviews were carried out with GPs. The results were recorded using keywords and evaluated in terms of content and topic. New topics were

identified in the first series of interviews (n = 9). A second series (n = 5) revealed no new topics, so we assumed that all relevant topics had been included. Based on these results, a questionnaire was developed that aimed to take all aspects into consideration, while being short enough to ensure a high response rate. The questionnaire was checked for comprehensibility by five GPs.

## Structure of the questionnaire

This analysis contains eight demographic items, 48 closed items (response scales: yes/no, yes/probably yes/probably no/no, very low/low/moderate/high/very high) and two items requiring GPs to provide exact numbers (e.g. "How many COVID-19 tests did you perform last week?"). The full questionnaire development is explained in the Supplement. The items not used in this paper will be analyzed in the longitudinal arm of the COVI-Prim study. Out of the 48 items used in this analysis eight factors were calculated. Reflecting the items contained within them, the factors were named as follows: (1) preparedness for a pandemic, (2) testing suspected cases, (3) protection of staff, (4) provision of information to GP, (5) perception of risk, (6) self-confidence, (7) decrease in number of patient contacts, (8) efforts to control the spread of the disease. Factor scores ranged from 0–10. The internal consistency (Cronbach's Alpha) of these eight factors used in this analysis ranged from $\alpha = .48$ to $\alpha = .85$ (S1 Table in S1 File).

## Survey

The questionnaire was transferred to LimeSurvey®. Invitations to GPs to respond to the questionnaire were sent out by participating universities in Austria (Graz, Salzburg, Innsbruck) and Germany (Frankfurt, Bochum, Hanover, Marburg, Gießen, Dresden, Freiburg, LMU Munich, Muenster, Aachen) using their respective mailing lists. Local GP associations, the Association of General Practitioners in Bavaria, Lower-Saxony and Baden-Wuerttemberg, Austria, and the Austrian Forum for Primary Care (OEFOP) also invited their members to participate. In accordance with data protection regulations, the study team did not have direct access to mailing lists. As the lists probably overlapped, it is not possible to know precisely how many GPs were contacted or to calculate a response rate. At the beginning of the survey, participants received information about its length, the investigators, and the purpose of the study. After ending the survey, all data on the online platform was stored in SPSS files. GPs received no incentive to participate.

## Statistics

Baseline characteristics are presented as mean ±SD or median (min-max), as appropriate. Categorical variables are provided as absolute numbers and in percent. In the main analysis, environmental variables (country of survey: Germany vs. Austria; size of town of practice: $< 5,000$ vs. 5,000 - $<20,000$ vs. 20,000 - $<100,000$ vs. $\geq 100,000$; type of practice: single-handed vs. not single handed) that may have influenced the responses were analyzed using General Linear Models. The main effects and all two-way interactions were therefore analyzed. Bonferroni correction was used to take account of multiple testing. Estimated means and 95% confidence intervals were used to present the results. For a better understanding of the results, responses to the items of the factors were also presented. In this presentation, the response categories "yes" and "probably yes" and the response categories "probably no" and "no" were combined. Items that did not belong to a factor were analyzed using ordinal regression analysis. No statistical correction was carried out to adjust for non-representative samples.

### Ethics

The study protocol has been approved by the local ethics committee of Goethe University Frankfurt, Germany (20–619). According to the Austrian law, this study does not require an ethical approval.

## Results

### Demographics

The survey was answered by 2,187 Austrian and German GPs during the early phase of the COVID-19-pandemic (3rd April to 30th April). The majority of GPs were male (55.6%), practiced in a city with fewer than 20,000 inhabitants (59.4%) and had a single-handed practice (57.7%). Mean age of the GPs was 52.5 years (SD: 9.6). In the week prior to answering the questionnaire, 56.1% of the GPs (n = 1226) ordered at least one COVID-19 test. In total 13,520 tests were ordered. Of the 1,226 GPs that ordered COVID-19 tests, 41.0% (n = 503; 41 GPs did not answer the question on the test results) received positive results for 1,593 patients (12.1% of 13,139 tests; 12.1%). All demographic characteristics are provided in Table 1.

### Overall results

GPs gave low ratings to their preparedness for a pandemic (mean: 2.7; 95% CI: 2.5–2.8, n = 1989), testing of suspected cases (3.3, 95%CI 3.2–3.4) and efforts to protect staff (2.0 95% CI 1.9–2.1). The provision of information to GPs (4.3, 95%CI: 4.2–4.4) and the perception of risk (5.1 95%CI 4.9–5.2) were rated as moderate. On the other hand, the participants rated their self-confidence (7.7, 95%CI 7.5–7.8), a decrease in patient contacts (6.8, 95%CI 6.7–7.0) and their efforts to control the spread of the disease (7.3, 95%CI 7.2–7.4) highly.

### Pandemic preparedness

Looking back to the beginning of the pandemic, 88.2% of GPs said they did not have enough protective equipment and 91.4% stated that they did not receive sufficient information on how much protective equipment they needed. Furthermore, a substantial number of GPs did not know where to procure protective equipment (78.3%) and said their practice was not well prepared for the COVID-19 pandemic (77.2%).

### Testing of suspected cases

Of the participants, 92.5% agreed that GPs should decide which patients should undergo testing for COVID-19. The idea of a telephone hotline for the exclusive use of medical staff ordering COVID-19 tests was approved by 86.9% of respondents. Of the GPs, 83.6% rejected the idea that all suspected cases of COVID-19 should be sent directly to hospital to enable them to focus on other patients. Furthermore, a large number of GPs said too little testing is performed (71.9%) and that they did not have adequate access to tests at the beginning of the pandemic (71.0%).

### Decrease in patient contacts

Of the GPs, 95.2% had less contact to patients as a result of the pandemic. Of these, 71.9% said they had less workload at the time because many patients are avoiding coming to the practice.

**Table 1. Baseline demographics.**

| | All | Germany | Austria |
|---|---|---|---|
| | n = 2187 | n = 1287 | n = 900 |
| Age (years) | 52.2 ± 9.6 | 51.7 ± 9.5 | 53.8 ± 9.6 |
| Sex | | | |
| male | 1217 (55.6%) | 673 (52.3%) | 544 (60.4%) |
| female | 965 (44.1%) | 609 (47.3%) | 356 (39.6%) |
| other | 5 (0.2%) | 5 (0.4%) | 0 (0.0%) |
| Size of town of practice | | | |
| < 5,000 | 658 (30.1%) | 264 (20.5%) | 394 (43.8%) |
| 5,000 - <20,000 | 642 (29.4%) | 421 (32.7%) | 221 (24.6%) |
| 20,000 - <100,000 | 635 (16.1%) | 287 (22.3%) | 66 (7.3%) |
| ≥100,000 | 534 (24.4%) | 315 (24.5%) | 219 (24.3%) |
| Type of practice | | | |
| single-handed | 1262 (57.7%) | 505 (39.2%) | 757 (84.1%) |
| not single-handed | 952 (42.3%) | 782 (60.8%) | 143 (15.9%) |
| Position in the practice | | | |
| employed | 213 (9.7%) | 202 (15.7%) | 11 (1.2%) |
| owner | 1945 (88.9%) | 1080 (83.9%) | 865 (96.1%) |
| locum | 29 (1.3%) | 5 (0.4%) | 24 (2.7%) |
| Year practice was established | median: 2003 | 2005 | 2003 |
| | Range: 1975–2020 | 1975–2020 | 1975–2020 |
| GPs that ordered COVID-19 tests in previous 7 days | | | |
| no | 760 (34.8%) | 289 (22.5%) | 471 (52.3%) |
| yes | 1226 (56.1%) | 916 (71.2%) | 310 (34.4%) |
| missing | 201 (9.2%) | 82 (6.4%) | 119 (13.2%) |
| GPs with patients with positive COVID-19 test results in previous 7 days | | | |
| (n = 1226) | | | |
| no | 682 (55.6%) | 520 (56.8%) | 162 (52.3%) |
| yes | 503 (41.0%) | 368 (40.2%) | 135 (43.5%) |
| missing | 41 (3.3%) | 28 (3.1%) | 13 (4.1%) |

Of the 2,187 GPs, 1,989 (90.9%) rated enough items to be included in the analysis. The median time required to answer the questionnaire was 14.1 minutes (IQR: 10.5–20.2 minutes) in Austria and 13.4 minutes (IQR: 9.8–19.0) in Germany. The completion rate of the survey was 79.7% in Austria and 85.2% in Germany.

## Information

Of the GPs, 71.4% said they had received insufficient information from public bodies. Before officially informing GPs of new developments, public authorities distributed important information to the general public via the media (70.9%).

## Self-confidence

Almost all the GPs said they knew what to do in suspected cases of COVID-19 (99.1%), and 82.1% were convinced they knew enough to provide optimal care for their patients during the pandemic.

## Efforts to control the spread of the virus in the practice

Almost all GPs tried to gain enough information from patients by phone beforehand to know whether they were dealing with a suspected case of COVID-19 (98.5%), and they took

precautions to ensure that suspected cases did not come into contact with other patients in their practice (97.4%). Over 80% of GPs avoided treating patients with mild symptoms that were not clearly linked to suspected cases of COVID-19 in their practice and preferred to attend to them by phone or online (87.9%). The distribution of responses is given in S1 Table.

## Economic aspects

60.0% of GP were concerned about how the pandemic would affect their own and their employees' economic prospects.

## Regional differences

Differences in the GP's responses were found to depend on the country in which the survey was conducted, the size of the city in which the practice was located and whether the practice was single-handed or not. No interactions between observed variables were significant.

Compared to Austrian GPs, German GPs rated their self-confidence lower (Germany: 7.5 95%CI: 7.4–7.6 vs. Austria: 7.8 95%CI: 7.6–8.0; p = .009), as they did their efforts to control the spread of SARS-CoV-2 (Germany: 7.1 95%CI: 7.0–7.2 vs. Austria: 7.5 95%CI: 7.3–7.6; p = .001). However, they rated their testing of suspected cases higher (Germany: 4.0 95%CI: 3.9–4.2 vs. Austria: 2.5 95%CI: 2.3–2.7; p = .009) and were more likely to say the number of patient contacts had decreased (Germany: 7.1 95%CI: 7.0–7.1 vs. Austria: 6.6 95%CI: 6.4–6.8; p < .001) (Table 2, Fig 1). Looking at single items, the biggest difference between German and

**Table 2. Mean and 95%CI for each factor of the evaluation of the pandemic for the whole group and subgroups.**

| | overall | Type of practice (single-handed) | | Country of survey | | City size | | | |
| | | yes | no | Austria | Germany | <5,000 | 5,000 –<20,000 | 20,000 –<100,000 | ≥ 100,000 |
|---|---|---|---|---|---|---|---|---|---|
| Preparedness for | 2.7 | 2.5 | 2.8 | 2.6 | 2.7 | 2.7 | 2.5 | 2.8 | 2.6 |
| a pandemic | (2.5–2.8) | (2.4–2.7) | (2.6–3.0) | (2.4–2.8) | (2.6–2.9) | (2.5–2.8) | (2.4–2.7) | (2.5–3.1) | (2.4–2.8) |
| Testing of | 3.3 | 3.2 | 3.3 | 2.5 | 4.0* | 3.2 | 3.3 | 3.4 | 3.2 |
| suspected cases | (3.2–3.4) | (3.1–3.3) | (3.2–3.5) | (2.3–2.7) | (3.9–4.2) | (3.1–3.4) | (3.1–3.4) | (3.1–3.6) | (3.0–3.4) |
| Protection of staff | 2 | 1.8 | 2.2 | 2.1 | 1.9 | 2 | 2.1 | 1.9 | 2 |
| | (1.9–2.1) | (1.7–2.0) | (2.0–2.4) | (1.9–2.4) | (1.7–2.0) | (1.8–2.2) | (1.9–2.3) | (1.6–2.2) | (1.8–2.2) |
| Provision of | 4.3 | 4.3 | 4.3 | 4.2 | 4.5 | 4.4 | 4.3 | 4.4 | 4.2 |
| information to GPs | (4.2–4.4) | (4.2–4.5) | (4.1–4.5) | (3.9–4.4) | (4.3–4.6) | (4.2–4.6) | (4.1–4.5) | (4.1–4.7) | (3.9–4.4) |
| Perception of risk | 5.1 | 5 | 5.1 | 4.8 | 5.3 | 5.1 | 5 | 5.1 | 5 |
| | (4.9–5.2) | (4.8–5.2) | (4.9–5.4) | (4.6–5.1) | (5.1–5.4) | (4.9–5.4) | (4.8–5.3) | (4.7–5.5) | (4.8–5.3) |
| Self-confidence | 7.7 | 7.7 | 7.6 | 7.8 | 7.5* | 7.6† | 7.6 | 8.0† | 7.4‡,§ |
| | (7.5–7.8) | (7.5–7.7) | (7.5–7.8) | (7.6–8.0) | (7.4–7.6) | (7.5–7.8) | (7.5–7.8) | (7.7–8.2) | (7.2–7.5) |
| Decrease in number | 6.8 | 6.9 | 6.8 | 6.6 | 7.1* | 6.7 | 6.7 | 7 | 6.9 |
| of patient contacts | (6.7–7.0) | (6.8–7.1) | (6.5–7.0) | (6.4–6.8) | (7.0–7.2) | (6.5–6.9) | (6.6–6.9) | (6.7–7.3) | (6.7–7.1) |
| Efforts to control | 7.3 | 7.2 | 7.4 | 7.5 | 7.1* | 7.3 | 7.3 | 7.2 | 7.3 |
| the spread of the | (7.2–7.4) | (7.1–7.3) | (7.3–7.6) | (7.3–7.6) | (7.0–7.2) | (7.2–7.5) | (7.2–7.5) | (7.0–7.4) | (7.2–7.5) |
| disease in the practice | | | | | | | | | |

Significant differences are in bold. (Scale values range from 0–10)

* Comparison Austria vs. Germany, p < .05

† . . . Variable city size: Post Hoc comparison to ≥ 100,000, p < .05 (Bonferroni corrected)

‡ . . . Variable city size: Post Hoc comparison to <5,000, p < .05 (Bonferroni corrected)

§ . . . Variable city size: Post Hoc comparison to 20,000 - <100,000, p < .05 (Bonferroni corrected)

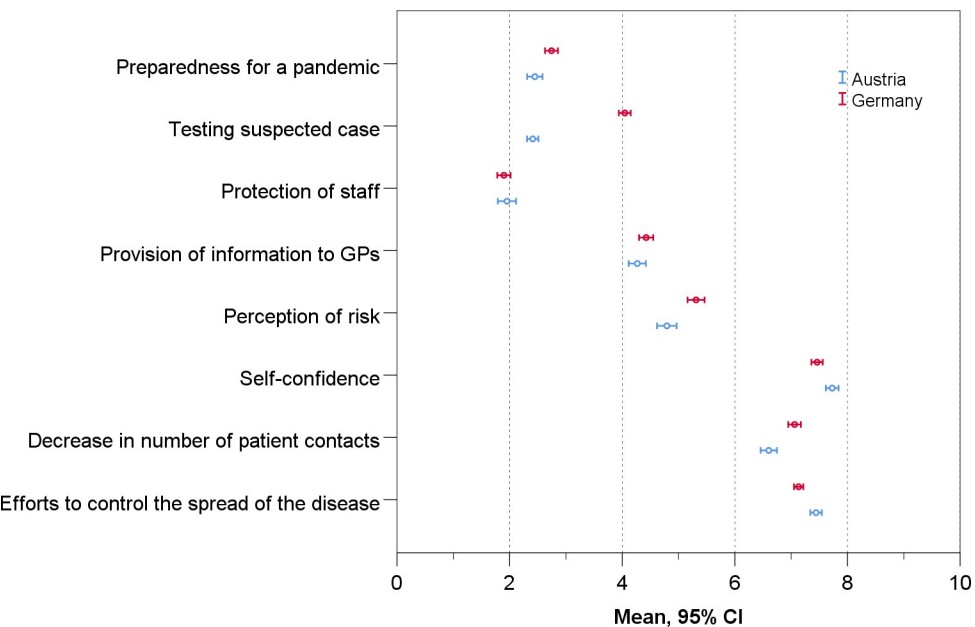

**Fig 1. Differences between German and Austrian GPs in their evaluation of the pandemic (Austria: n = 900; Germany: n = 1287).**

Austrian GPs was found in testing, with 62.8% of German GPs saying too little testing was carried out, compared to 84.9% of Austrian GPs, and 42.4% of German GPs saying they had adequate access to tests at the beginning of the pandemic, compared to 9.7% of Austrian GPs. Regarding the items that did not belong to an factor the following differences between Austrian and German GPs were observed. Austrian GPs were less worried about how the pandemic will affect their economic situation (p < .001), kept a close eye on themselves and their employees to see whether anyone was showing initial symptoms of an infection (p = .002) and more often had to take on patients from colleagues that had closed their practice because of quarantine (p < .001) (S3 Table).

While no differences in factor score were found between GPs working in single-handed practice or not did, responses to items, that did not belong to a factor were observe. GPs working in single-handed practice were more worried about how the pandemic will affect their economic situation (p = .018), kept a close eye on themselves and their employees to see whether anyone was showing initial symptoms of an infection (p < .001), had less sufficient information on the type of personal protective equipment (p = .017) and had not so often take on patients from colleagues that had closed their practice because of quarantine (p = .046) (S3 Table).

GPs in cities with 100,000 inhabitants or more rated their self-confidence lower than GPs in towns with fewer than 5,000 (p = .041) and towns with 20,000–100,000 (p < .001) inhabitants (Fig 2, Table 2). Analyzing the items used to calculate the self-confidence score, the largest difference can be observed in GPs' conviction that their knowledge was sufficient to provide optimal care for their patients during the pandemic. While 87.1% of GPs in cities with 20,000–100,000 inhabitants were convinced, the number fell to 82.9% in cities with fewer than 5,000 inhabitants and to 79.0% in cities with 100,000 or more inhabitants. Regarding the items that did not belong to a factor the following differences were observed. GPs in cities with 100,000 or more inhabitants were more worried about how the pandemic will affect their economic situation (p = .001) and more often had to take on patients from colleagues that had

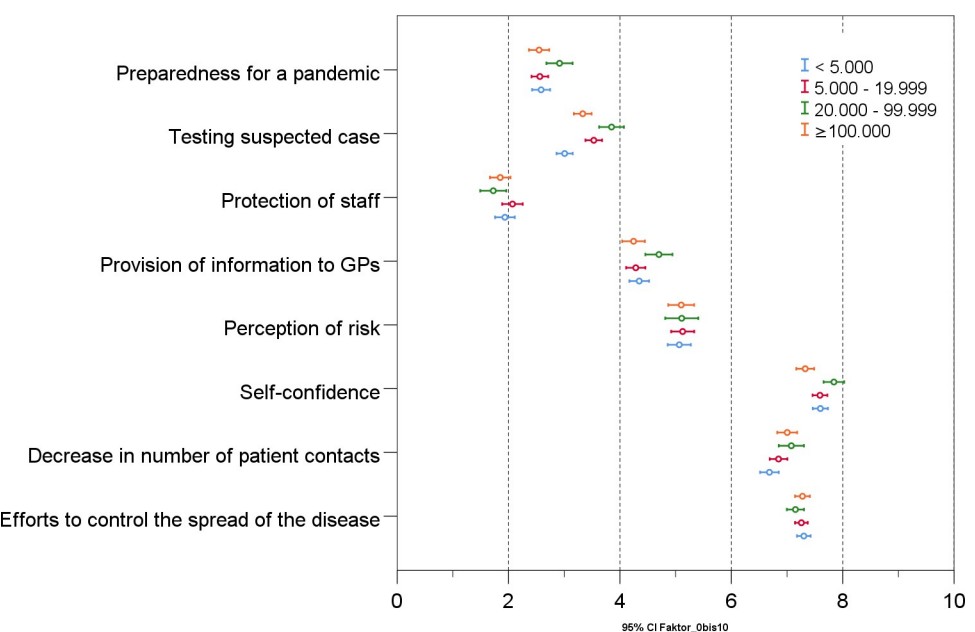

**Fig 2. Differences in the evaluation of the pandemic of GPs with practices in cities of different sizes.**

closed their practice because of quarantine (p = .003) compared to GPs in towns with fewer than 5,000 (S3 Table).

## Discussion

Our survey covered the specific problems and experiences of more than two thousand general practitioners in Austria and Germany at the beginning of the COVID-19 pandemic. The high level of participation demonstrates the interest and concern of this group. In the early stages, GP practices were not well prepared and did not have enough protective equipment. GPs did not receive sufficient information from public stakeholders but were very active on informal digital networks involving their professional peer group. Overall, they had fewer patient contacts. A majority wanted to decide themselves whom to test, and to have a higher number of tests made available to GPs themselves. They were concerned about the economic outlook but they were generally self-confident in terms of dealing with suspected and confirmed cases of COVID-19.

Considering its scale and abruptness, the reported lack of preparation for an event such as the COVID-19 pandemic is not surprising. Even though GPs immediately went to great lengths to procure enough protective equipment and to re-organize and adapt standard procedures in their practices, some–as in other countries–also had to work without sufficient PPE [17–19]. Since the availability of PPE is essential to ensuring the continuous and safe provision of care during a pandemic, it is critical to incorporate primary care practices in the procurement of PPE. Existing structures should support the development of a joint national response plan to ensure that primary care is adequately involved [10].

Although many of the challenges such as that mentioned above were observed internationally, some regional differences stand out. In particular, GPs in Austria were not initially involved in testing procedures. Instead, the population in Austria was encouraged to contact an official health hotline in case of symptoms or suspicion of infection. Hence, GPs were overlooked in their role as gatekeepers in primary care. For GPs, this is likely to have been

particularly frustrating, as the vast majority are convinced they know how to manage patients with a suspected infection and are willing to do so.

Furthermore, in the current situation it is especially important to motivate primary care practitioners, as they are in the frontline in terms of contact with the community [3]. The role of the GP is to decide which patients need hospital care and to monitor others at home [20]. This is the only way to ensure that important resources in hospitals are not overburdened. Experts' concerns that a significant number of patients may die or suffer harm due to delayed access to usual medical care [21, 22] are also important and are reflected in our survey. As noted above, the number of patients visiting primary care practices decreased during the COVID-19 pandemic. People had strict stay-at-home orders or were afraid of infection. However, a few weeks after the lockdown, there was widespread criticism that this may have led to significant collateral damage. Several recently published articles pointed out that fewer patients were diagnosed with serious medical conditions such as stroke [23], acute coronary syndrome [24], atrial fibrillation [25] and cancer [26]. Furthermore, the WHO warned that measures designed to slow the spread of the coronavirus might also delay vaccination programs and thereby speed up the spread of other vaccine-preventable diseases [27].

General practitioners are responsible for the population as a whole, and the COVID-19 pandemic affected everyone. While children usually only experience mild or asymptomatic disease symptoms [28], they are also strongly affected by social isolation. A lack of structure and support from schools can increase anxiety and potentially impact mental health [29]. Other vulnerable groups to consider are elderly people that are living alone and for whom the use of online communication systems is often not feasible, as well as those with mental health problems, or people living in poor socio-economic conditions. They are all part of the patient collective in a primary care setting. We therefore need strategies to avoid future collateral damage that ensure access to primary care, even at times of high infection rates. Possible solutions, such as the greater use of telemedicine appointments and triage for certain patient groups according to the severity and urgency of a consultation, are surveyed in our longitudinal study (see supporting information), for which the analysis is ongoing.

But telemedicine alone is not enough. About 60% of GPs reported financial and economic concerns. This suggests that existing remuneration mechanisms for primary care need to be adapted or amended during a pandemic. Basu et al. estimated that the losses to primary care practices resulting from the pandemic amounted to about 15 billion USD in the U.S. alone [30].While SARS-CoV-2 is certainly the most serious pandemic since the influenza pandemic of 1917–18 [31], it has not been the only one in recent years. The H1N1 virus in 2009 was also declared responsible for an influenza pandemic and resulted in widespread preparations. However, it had far less impact on the population than expected, and a specific vaccine and treatment was available early [32]. SARS-CoV-1 in 2003 resulted in a similar public health response in strongly affected regions like Toronto [33]. Many of the issues that arose during that outbreak are mirrored in this pandemic on a global scale and can be found in the results of our study. Such pandemics, as well as seasonal influenza epidemics, lead to a surge in hospital bed demand and primary care consultations [34]. The COVID-19 pandemic is somewhat different because a strong focus was placed on saving health care resources in countries that had time to prepare before the need for them had arisen.

Our study has some limitations. Firstly, the questionnaire was developed in a very short time so that it could be delivered when the situation was most acute. Even though we tried to include all relevant topics, some issues may have been missed. Secondly, we could not calculate the response rate because a systematic area-wide survey was not possible in the time frame we permitted ourselves. However, the number of responses far exceeded our expectations, especially considering the difficulties that are usually encountered in recruiting GPs for research

projects [35]. In addition, the questionnaire was completed by a very high percentage of participants. Thirdly, the recruitment process through regional networks and professional associations led to the heterogeneous selection of participants, which may have limited representativeness. One further limitation is that our survey was only carried out among GPs and did not involve other team members from the primary care setting.

Primary care is an important and vital resource for dealing with a pandemic like COVID-19. The workforce is confident and willing to take an active role, but needs to be given the opportunity and provided with the necessary conditions to do so. As GPs work on the frontline, they should be adequately supported, both in terms of the provision of protective equipment and financial security during the active phase of the pandemic. To ensure a quick and effective response to any new crisis, general practitioners in primary care should be involved in a national coordinated strategy that includes all relevant parties.

## Supporting information

**S1 Fig. Differences between GPs in single-handed and not single-handed practices in their evaluation of the pandemic.**
(DOCX)

**S1 Table. Response distribution (%) for all items.**
(DOCX)

**S2 Table. Difference in the responses of Austrian and German GPs.** Percentages were calculated as %German GPs minus %Austrian GPs. Responses which were more often chosen by German GPs are marked green and responses which were more often chosen by Austrian GPs are marked red.
(DOCX)

**S3 Table. Differences in responses to items that do not belong to a factor (multivariable ordinal or binary logistic regression results; Bonferroni correction).**
(DOCX)

**S1 File. Project description.** Questionnaire development, Structure of the Questionnaires, Translation, Survey, Statistics, Ethics.
(DOCX)

**S2 File. Checklist for Reporting Results of Internet E-Surveys (CHERRIES).**
(DOCX)

**S3 File. COVI Prim Baseline questionnaire—English.**
(DOCX)

**S4 File. COVI Prim Baseline questionnaire–German.**
(DOCX)

## Acknowledgments

We would like to thank all participating general practitioners, and the institutions that were willing to send the link to our questionnaire to their network partners.

## Author Contributions

**Conceptualization:** Andrea Siebenhofer, Sebastian Huter, Alexander Avian, Karola Mergenthal, Dagmar Schaffler-Schaden, Ulrike Spary-Kainz, Herbert Bachler, Maria Flamm.

**Data curation:** Sebastian Huter, Alexander Avian.

**Formal analysis:** Alexander Avian.

**Investigation:** Andrea Siebenhofer, Sebastian Huter, Alexander Avian, Karola Mergenthal, Dagmar Schaffler-Schaden, Ulrike Spary-Kainz, Herbert Bachler, Maria Flamm.

**Methodology:** Andrea Siebenhofer, Sebastian Huter, Alexander Avian, Karola Mergenthal, Dagmar Schaffler-Schaden, Ulrike Spary-Kainz, Herbert Bachler, Maria Flamm.

**Project administration:** Andrea Siebenhofer, Sebastian Huter, Karola Mergenthal, Dagmar Schaffler-Schaden, Maria Flamm.

**Resources:** Andrea Siebenhofer, Karola Mergenthal.

**Supervision:** Andrea Siebenhofer, Maria Flamm.

**Visualization:** Alexander Avian.

**Writing – original draft:** Andrea Siebenhofer, Sebastian Huter, Alexander Avian, Karola Mergenthal, Dagmar Schaffler-Schaden, Ulrike Spary-Kainz, Herbert Bachler, Maria Flamm.

**Writing – review & editing:** Andrea Siebenhofer, Sebastian Huter, Alexander Avian, Karola Mergenthal, Dagmar Schaffler-Schaden, Ulrike Spary-Kainz, Herbert Bachler, Maria Flamm.

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
