## [Editor Report · Decision Letter 0]

5 Feb 2021

PONE-D-20-36154

COVI-Prim survey: Challenges for Austrian and German general practitioners during initial phase of COVID-19

PLOS ONE

Dear Dr. Avian,

Thank you for submitting your manuscript to PLOS ONE. After careful consideration, we feel that it has merit but does not fully meet PLOS ONE’s publication criteria as it currently stands. Therefore, we invite you to submit a revised version of the manuscript that addresses the points raised during the review process.

This is a good study focused on “Challenges for Austrian and German general practitioners during

initial phase of COVID-19”. Please address the following points before external review:

• You must provide a rational for absence of Ethics approval for Austria as mentioned in appendix table 1 considered as Not Applicable.

• The result of open-end question (free text in questionnaire) was not presented in manuscript. It appears the most related part of questionnaire to your manuscript title was this section, “biggest challenge, supporting factor”.

• The results of last table in questionnaire were not presented at all. And were not analyzed in relation to other factors.

• Considering the unity of covi-prim project please clarify your rational to publish the result of this project Separately.

We look forward to receiving your revised manuscript.

Kind regards,

Kamal Gholipour, PhD

Academic Editor

PLOS ONE

Journal Requirements:

"The study was financed by the cooperating University Institutes without any external financial

130 support"

"The authors received no specific funding for this work."
---

## [Author Response · Author response to Decision Letter 0]

22 Feb 2021

• You must provide a rational for absence of Ethics approval for Austria as mentioned in appendix table 1 considered as Not Applicable.

Response: When we were asking the head of the local ethic committee at the Medical University Graz for an ethical approval, we were told that according to the Austrian law, this study does not require an ethical approval. This information will be added in the method section in the paragraph Ethics at page 6 and the approval from Germany will be uploaded as a supplement file. 

• The result of open-end question (free text in questionnaire) was not presented in manuscript. It appears the most related part of questionnaire to your manuscript title was this section, “biggest challenge, supporting factor”.

Response: Prior to the writing of our manuscript we were discussing this aspect in detail and decided that we do not include the comments. This decision was based on the high number (more than 3.500 comments) and the heterogeneity of the comments. We observed that comments differed over time and were strongly dependent of the region of the responding doctors. In our opinion, it is not possible to report these results in one paragraph. We are currently preparing a second manuscript reporting on the interesting results to present the comments in detail. This information can be added in our actual manuscript for PLOS ONE.

• The results of last table in questionnaire were not presented at all. And were not analyzed in relation to other factors.

Response: We have not analyzed this items in the first version of the manuscript, because, these items do not belong to one of the dimensions of the questionnaire. However, based on your suggestion, we now added the most important results in the manuscript and the other results in the appendix (table S5)

• Considering the unity of covi-prim project please clarify your rational to publish the result of this project separately.

Response: Considering the number of different aspects of the COVI-Prim project, we think that it is not possible to present all results within one manuscript. There are so many aspects (changes over time, differences between countries, differences in baseline between GPs answering the first time at the beginning of the pandemic and GPs answering at the beginning of the summer, a.s.o.) that each of these aspect could be described in several further manuscripts. Therefore, we decided to write one “starter” manuscript including all protocol relevant information in the main text and the supplement that we would be able to reference it within every subsequent publication.

---

## [Decision Letter · Decision Letter 1]

3 May 2021

COVI-Prim survey: Challenges for Austrian and German general practitioners during initial phase of COVID-19

PONE-D-20-36154R1

Dear Dr. Avian,

We’re pleased to inform you that your manuscript has been judged scientifically suitable for publication and will be formally accepted for publication once it meets all outstanding technical requirements.

Kind regards,

Kamal Gholipour, PhD

Academic Editor

PLOS ONE

Additional Editor Comments (optional):

Reviewers' comments:

Reviewer's Responses to Questions

**Comments to the Author**

1. If the authors have adequately addressed your comments raised in a previous round of review and you feel that this manuscript is now acceptable for publication, you may indicate that here to bypass the “Comments to the Author” section, enter your conflict of interest statement in the “Confidential to Editor” section, and submit your "Accept" recommendation.

Reviewer #1: (No Response)

2. Is the manuscript technically sound, and do the data support the conclusions?

Reviewer #1: Yes

3. Has the statistical analysis been performed appropriately and rigorously? 

Reviewer #1: Yes

4. Have the authors made all data underlying the findings in their manuscript fully available?

Reviewer #1: Yes

5. Is the manuscript presented in an intelligible fashion and written in standard English?

Reviewer #1: Yes

6. Review Comments to the Author

Reviewer #1: The introduction and title are well written and the aim of the study is clear with an obvious gap of knowledge.

The methods is well written and reproducible

In the results I recommend investigating the association between level of knowledge and experience or grades in medical school. Otherwise the results are well represented

The discussion is well written however I recommend comparing the results with other studies published on this topic which are numerous and regional.

7. PLOS authors have the option to publish the peer review history of their article (what does this mean?). If published, this will include your full peer review and any attached files.

Reviewer #1: **Yes: **Kirellos Said Abbas

---

## [Editor Report · Acceptance letter]

17 May 2021

PONE-D-20-36154R1 

COVI-Prim survey: Challenges for Austrian and German general practitioners during initial phase of COVID-19 

Dear Dr. Avian:

I'm pleased to inform you that your manuscript has been deemed suitable for publication in PLOS ONE. Congratulations! Your manuscript is now with our production department. 

Kind regards, 

on behalf of

Dr. Kamal Gholipour 

Academic Editor

PLOS ONE